# Research on Polyp Segmentation via Dynamic Multi-Scale Feature Fusion and Global–Local Semantic Enhancement

**DOI:** 10.3390/s25206495

**Published:** 2025-10-21

**Authors:** Wei Qing, Yuyao Ouyang, Pengfei Yin

**Affiliations:** College of Computer Science and Engineering, Jishou University, Jishou 416000, China; 2023404585@stu.jsu.edu.cn (W.Q.); 2023402786@stu.jsu.edu.cn (Y.O.)

**Keywords:** colon polyp segmentation, YOLOv12-seg, multi-scale feature fusion

## Abstract

Accurate segmentation of colorectal polyps is crucial for the early screening and clinical diagnosis of colorectal cancer. However, the diverse morphology of polyps, significant variations in scale, and unstable quality of endoscopic imaging pose serious challenges for existing algorithms in achieving precise boundary segmentation. To address these issues, this study proposes a novel polyp segmentation algorithm, GDCA-Net, which is developed based on the You Only Look Once version 12 segmentation model (YOLOv12-seg). GDCA-Net introduces several architectural innovations. First, a Gather-and-Distribute (GD) mechanism is incorporated to optimize multi-scale feature fusion, while Alterable Kernel Convolution (AKConv) is integrated to enhance the modeling of complex geometric structures. Second, the Convolution and Attention Fusion Module (CAF) and Context-Mixing dynamic convolution (ContMix) modules are designed to strengthen long-range dependency modeling and multi-scale feature extraction for polyp boundary representation. Finally, a Wise Intersection over Union–based (Wise-IoU) loss function is introduced to accelerate model convergence and improve robustness to low-quality samples. Experiments conducted on the PolypDB, Kvasir-SEG, and CVC-ClinicDB datasets demonstrate the superior performance of GDCA-Net in polyp segmentation tasks. On the most challenging PolypDB dataset, GDCA-Net achieved a mean Average Precision at 50% IoU threshold (mAP50) of 85.9% and an F1-score (F1) of 85.5%, representing improvements of 2.2% and 0.7% over YOLOv12-seg, respectively. Moreover, on the Kvasir-SEG dataset, GDCA-Net achieved a leading F1 score of 94.9%. These results clearly demonstrate that GDCA-Net possesses strong performance and generalization capabilities in handling polyps of varying sizes, shapes, and imaging qualities.

## 1. Introduction

Accurate segmentation of colorectal polyps plays a critical role in computer-aided medical diagnosis, facilitating early detection, timely intervention, and effective prevention of malignant transformation [1]. However, existing segmentation algorithms still face considerable challenges in real clinical applications.

First, under multimodal imaging conditions, polyps exhibit significant variations in color, texture, boundary features, morphology, and scale, making it difficult for conventional pixel-level feature extraction methods to capture fine-grained information [2]. Second, many existing approaches tend to overemphasize local features while neglecting the complex interactions between the global semantic context and local structural details. As a result, their performance often deteriorates when dealing with blurred boundaries or small, low-contrast polyps. In addition, insufficient modeling of long-range dependencies and high sensitivity to low-quality inputs such as noise and specular reflections further undermine robustness and generalization, thereby limiting their clinical utility in fine-grained segmentation tasks [3].

In recent years, the rapid advancement of deep learning has brought about significant breakthroughs in medical image segmentation [4,5,6,7]. Many studies have demonstrated promising results by introducing attention mechanisms [8], convolutional kernel modeling [9,10], and context-aware feature representations [11]. Nevertheless, these methods still face limitations in balancing global and local feature integration, adapting to complex morphological variations, and maintaining robustness against low-quality imaging data, which restricts their widespread clinical adoption [12].

To address these challenges, this study proposes a novel deep neural network architecture that achieves superior segmentation performance and enhanced robustness through a multi-module collaborative optimization strategy. Specifically, to tackle the difficulty of jointly modeling local details and global context in fine-grained boundary segmentation, we designed the CAFMAttention mechanism [13], which simultaneously captures local detail features and global contextual information, thereby improving boundary delineation. To overcome the limitations of conventional convolution in modeling long-range dependencies, we introduce the ContMix module [14], a context-aware dynamic convolution mechanism that breaks the locality constraint of standard convolution. This module adaptively models long-range dependencies, thereby significantly enhancing feature representation and improving model generalization.

Furthermore, to address the inherent information loss in traditional feature pyramid networks (FPNs), we propose a Gather-and-Distribute (GD) mechanism [15], which collaboratively optimizes feature utilization efficiency and strengthens the detection capability of each branch. We also employ Arbitrary Kernel Convolution (AKConv) [16], a dynamic convolution module capable of generating and adjusting kernel sampling coordinates. This enhances the modeling of multi-scale and irregular polyp structures, thereby improving small-object detection performance.

Finally, to mitigate the sensitivity to low-quality samples and slow convergence in bounding-box regression, we adopt the improved Wise-IoU loss function [17]. By integrating a dual-attention mechanism with a dynamic gradient gain strategy, this loss function adaptively evaluates anchor-box quality and allocates gradients, thereby reducing sensitivity to low-quality samples, accelerating convergence, and enhancing robustness across datasets of varying quality.

The main contributions of this study can be summarized as follows:We propose a multiscale feature fusion framework based on the GD mechanism and AKConv. The GD module integrates three key components: the Feature Alignment Module (FAM), the Information Fusion Module (IFM), and the Information Injection Module (Inject). Together with AKConv, this design significantly enhances branch segmentation capability and feature utilization, improving the modeling of complex geometric structures.We design a local–global semantic extraction mechanism based on CAFM and ContMix. This approach adaptively generates convolutional kernels from input features, enabling effective modeling of long-range dependencies and significantly improving feature representation.We introduce Wise-IoU, a loss function that combines a dual-attention mechanism with a dynamic gradient gain strategy. This loss accelerates model convergence and enhances adaptability to samples of varying quality.

## 2. Related Works

Image-based polyp segmentation is an interdisciplinary research area that integrates image processing, pattern recognition, machine learning, and deep learning. Current approaches can be broadly divided into two categories: traditional methods based on hand-crafted feature engineering, where features such as shape, texture, and color are designed manually to achieve segmentation. These approaches are theoretically mature but suffer from limited generalizability. The other category comprises deep learning-based end-to-end segmentation models (e.g., U-Net [18] and DeepLab [19]), which can automatically learn pixel-level features and demonstrate stronger robustness and accuracy in complex scenarios.

### 2.1. Traditional Polyp Segmentation Methods

In traditional polyp segmentation, researchers typically follow a rigorous workflow. First, image preprocessing is a critical initial step aimed at improving visual quality, reducing noise and glare interference, and enhancing the distinguishability between polyps and surrounding normal tissues. For instance, image smoothing is often applied to suppress random noise [20]; histogram equalization redistributes pixel intensity values to enhance contrast, making otherwise indistinct polyps more visually separable from surrounding tissues. In addition, color-space transformations are frequently employed to emphasize the color characteristics of polyps [21]. The second key step is handcrafted feature extraction, whose performance largely depends on expert knowledge and a deep understanding of polyp morphology [22]. The goal is to design pixel-level descriptors that effectively discriminate between polyp and non-polyp regions. These features typically include shape descriptors that capture polyp geometry [23], texture features characterizing local grayscale variations [24], and boundary descriptors optimized using tools such as the Canny operator.

Finally, after extracting these handcrafted features, traditional methods typically employ machine learning algorithms for pixel-level classification [25]. Such approaches learn the mapping between feature vectors—constructed from handcrafted descriptors—and pixel categories (polyp vs. non-polyp), thereby enabling automatic segmentation of new images. Commonly used classifiers include support vector machines (SVMs), random forests, and decision trees [26]. The performance of these classifiers is highly dependent on the discriminative power of the extracted handcrafted features.

In summary, traditional polyp segmentation methods rely on meticulous image preprocessing [27,28], expert-driven handcrafted feature extraction, and classical machine learning classifiers. Although these approaches have achieved certain progress in the early stages, their performance is fundamentally limited by the representational power and generalizability of handcrafted features. Specifically, they struggle to capture the diverse morphology of polyps (e.g., flat vs. pedunculated types), their preprocessing pipelines often rely on manually tuned parameters with poor generalization, and classifiers based on low-level features exhibit inadequate accuracy when handling ambiguous boundaries. These limitations have driven the adoption of deep learning in medical image segmentation [29], accelerating the advancement of artificial intelligence-assisted colonoscopy.

### 2.2. Advanced Polyp Segmentation Methods

Traditional image processing and handcrafted feature engineering face notable challenges in segmenting polyps, especially for small or early-stage polyps with diverse morphology and texture. Consequently, current deep learning models in medical image segmentation actively explore strategies to effectively fuse local details with global contextual information and enhance the modeling of long-range dependencies.

Fully Convolutional Networks (FCNs) [30] and their variants (e.g., U-Net and DeepLab) extract hierarchical features using convolutional kernels and have become foundational and effective methods for medical image segmentation. However, due to the locality of convolutional receptive fields, these models struggle to capture long-range semantic dependencies [31]. To address this limitation, researchers have proposed the integration the local feature extraction capacity of CNNs with the global modeling capability of Transformers [32]. For instance, the Convolution-and-Attention Fusion Module (CAFM) was developed to strengthen both global and local feature modeling. Moreover, gated attention mechanisms have been explored to compensate for the lack of global semantic information in conventional convolutional operations.

In the area of dynamic convolution, OverLoCK proposed the Context-Mixing Dynamic Kernel (CMDK), which enhances feature representations through dynamic top-down attention. Furthermore, multi-scale feature fusion networks have opened new avenues for medical image segmentation. For example, UNet++ [33] employs densely connected multi-scale feature extraction paths to capture semantic information at different levels, representing a significant improvement over the classical U-Net. Similarly, HCANet [34] incorporates a Multi-Scale Feedforward Network (MSFN) with parallel dilated convolutions [35], which extracts features at varying receptive fields and markedly improves segmentation performance. Such multi-scale fusion is particularly important in medical image segmentation tasks.

In recent years, many studies have revisited and analyzed standard convolution operations from different perspectives, proposing novel convolutional operators to improve cross-scale recognition ability. For instance, Li et al. introduced the Involution operator, which reverses the feature mapping of convolution to enhance network performance. Zhang et al. [36] recognized that spatial attention essentially addresses the parameter-sharing problem of convolution and proposed RFAConv. Compared with these approaches, the AKConv adopted in our work provides more effective feature extraction.

Finally, to improve segmentation accuracy, it is essential to design a robust bounding-box regression (BBR) loss function that can effectively handle samples of varying quality. For instance, earlier works, such as GIoU [37] and DIoU [38], introduced modifications to the standard IoU loss. Focal-EIoUv1 was proposed to address this issue. However, due to its static focusing mechanism (FM), the potential of non-monotonic FM was not fully exploited. Moreover, FM-based loss functions exhibit a notable drawback: even simple samples produce relatively large loss values, thereby competing with hard samples. Lin et al. [39] introduced focal loss with a monotonic FM, which effectively reduces the competitiveness of easy samples. Subsequently, Zhang et al. [40] proposed Focal-EIoUv1 with non-monotonic FM and Focal-EIoU with monotonic FM. Their experiments demonstrated that monotonic FM outperformed its non-monotonic counterpart. Building upon these insights, we introduce Wise-IoU, a loss function with a dynamic non-monotonic FM. Unlike prior static mechanisms, Wise-IoU leverages regional outlierness to evaluate segmentation quality and adaptively allocate gradient gains. This enables the model to focus more on moderately difficult yet critical segmentation regions, thereby improving overall performance.

## 3. Methodology

### 3.1. YOLOv12-Seg Overview

YOLOv12-Seg is an advanced segmentation model developed on the basis of the YOLO family of architectures, with the design objective of balancing high-precision semantic modeling and real-time inference efficiency [41]. In medical image segmentation tasks, a model must not only delineate boundaries with high accuracy but also satisfy real-time requirements for clinical applications. Therefore, YOLOv12-Seg provides a feasible solution that integrates both speed and accuracy for medical image segmentation.

Structurally, YOLOv12-Seg inherits the backbone of YOLOv12 and incorporates improved convolutional and feature aggregation modules (C3k2 and A2C2f) to enhance multi-scale feature representation [42]. Through cross-stage connections and multi-level fusion [43], the model integrates semantic and spatial information across different layers. Its head includes a segmentation branch that jointly leverages P3, P4, and P5 feature maps for segmentation prediction, thereby preserving global contextual information while strengthening the delineation of edge details.

### 3.2. Improved Feature Fusion Network

Current YOLO algorithms typically adopt feature pyramid networks (FPNs) and PANet structures for multi-scale fusion [43]. However, the degree of fusion remains limited: the PAFPN used in the neck of the standard YOLO series can only fully integrate information between adjacent layers, while non-adjacent layers require recursive upward fusion. This not only increases complexity but also leads to potential information loss.

With the introduction of the GD mechanism, features at different scales are uniformly collected, summarized, and fused, after which the centralized information is redistributed to each level [15]. During this process, the Feature Alignment Module (FAM) and the Information Fusion Module (IFM) work jointly to achieve multi-level feature integration [44]. This approach enables the model to effectively leverage diverse features, thereby improving segmentation accuracy while maintaining low latency. It overcomes the information loss problem of conventional FPNs and strengthens the feature integration in the neck, making better use of the features extracted by the backbone.

As shown in Figure 1, the GD mechanism is integrated into the neck of the original YOLOv12-Seg network: Low-GD replaces the up-sampling fusion in PANet, while Higher-GD substitutes the down-sampling fusion. Additionally, higher-level B2 features are incorporated into Low-GD to maximize the integration of low-level information. Within Low-GD, the B4 layer is used as a reference: larger feature maps, such as those from B2 and B3, are down-sampled via average pooling, whereas smaller feature maps, such as those from B5, are up-sampled using bilinear interpolation to standardize feature-map sizes and obtain fused features. The fused feature layers (P3, P4, and P5) from Low-GD are subsequently processed by High-GD fusion, further enhancing information integration and effectively preserving the feature details of small objects.

In addition, ContMix is incorporated to adaptively adjust the shape and size of convolutional kernels, thereby enhancing flexibility in feature extraction. CAFMAttention is also introduced to effectively model long-range dependencies and strengthen local feature extraction while maintaining computational efficiency. Furthermore, AKConv is adopted, which accommodates arbitrary kernel shapes and sizes through flexible initial sampling and learnable offsets, thereby improving feature extraction capability and adaptability to objects with diverse shapes [45].

Finally, the improved Wise-IoU loss function is employed to optimize the model. By leveraging a dual-attention mechanism and dynamic gradient gain, Wise-IoU accelerates model convergence and enhances its adaptability to datasets of varying quality [46].

### 3.3. Principles of Information Aggregation and Distribution Mechanism

To effectively address the problem of information loss in traditional Feature Pyramid Networks (FPNs) during feature transmission, we introduce the Gather-and-Distribute (GD) mechanism, as shown in Figure 2. This mechanism employs a unified module to aggregate and fuse multi-level features, then redistributes the fused results back to each layer [47]. By doing so, it mitigates the inherent information loss of conventional methods while enhancing the feature integration capability of the neck, without introducing significant latency. Consequently, this approach enables more efficient utilization of backbone-extracted features and can be conveniently embedded into any existing backbone–neck–head architecture.

### 3.4. Convolution and Attention Fusion Module

To overcome the limitations of traditional convolutions in capturing long-range dependencies and global context, this project plans to introduce a Convolution and Attention Fusion Module (CAFM) [48], as shown in Figure 3. This module combines the strong local modeling capabilities of convolutions with the strong global perception capabilities of attention mechanisms, enabling more comprehensive extraction of multi-scale, multi-level semantic information and effectively enhancing the model’s feature expression capabilities and robustness in complex scenarios.

Unlike the standard Transformer self-attention mechanism, which computes global dependencies based on pairwise similarity among all tokens using Q (query), K (key), and V (value) matrices, the proposed CAFM introduces a dual-branch structure that explicitly combines local convolutional feature extraction with global attention modeling. The convolutional branch focuses on capturing fine-grained spatial and edge details with strong inductive bias, while the attention branch aggregates global context information without fully relying on pairwise token relationships.

### 3.5. Variably Sized Convolution

Convolution-based neural networks have achieved remarkable success in deep learning; however, standard convolution operations suffer from two inherent limitations. On the one hand, convolution is restricted to local receptive fields with fixed sampling patterns. On the other hand, the number of kernel parameters grows quadratically with kernel size. To address these limitations, this study investigates Alterable Kernel Convolution (AKConv), which enables convolution kernels to adopt arbitrary numbers of parameters and flexible sampling patterns, thereby offering richer trade-offs between computational cost and model performance. In AKConv, a novel coordinate generation algorithm is introduced to define the initial positions for kernels of arbitrary size, while offsets are employed to adjust the sampling pattern of each position.

#### 3.5.1. Define the Initial Sampling Position

Convolutional neural networks are based on convolution operations, which locate features at corresponding positions through a regular sampling grid. In [11,33,34], for a 3×3 convolution operation, the regular sampling grid is given by the following equation. Let R denote the sampling grid; then, R is defined as follows:(1)R={(−1,−1),(−1,0),⋯,(0,1),(1,1)}
Nevertheless, such a regular sampling strategy constrains the flexibility of kernel shapes. The AKConv explored in this work overcomes this limitation by allowing the convolution kernels to operate with irregular shapes. To provide irregular kernels with a structured sampling grid, we designed an algorithm capable of generating initial sampling coordinates for convolutions of arbitrary size, with the top-left corner of the kernel (0, 0) defined as the sampling origin to accommodate different kernel dimensions. After defining the initial coordinates ($P_{n}$) for an irregular convolution, the corresponding convolution operation at position $P_{0}$ can be defined as follows:(2)$$\mathrm{Conv}\left(P_{0}\right)=\sum w\times \left(P_{0}+P_{n}\right)$$
where *w* represents the weight parameters of the convolution kernel; $w\in R^{C\times K}$; where *C* is the number of channels, and *K* is the size of the convolution kernel.

#### 3.5.2. Novel Deformable Convolution Operation

Since the sampling positions of standard convolution are fixed, it can only capture local window information and fails to extract features from other regions. Although deformable convolution enhances flexibility by learning offsets to adjust sampling positions, it remains constrained to regular grids, and the number of kernel parameters increases quadratically with kernel size, leading to high computational overhead. To address these limitations, we propose a novel deformable convolution operation, termed AKConv, which enables convolutional kernels to adopt arbitrary sampling patterns and parameter counts, thereby adapting to targets of varying scales and irregular shapes. To address these limitations, we propose a novel deformable convolution operation, termed AKConv, which enables convolutional kernels to adopt arbitrary sampling patterns and parameter counts, thereby adapting to targets of varying scales and irregular shapes. However, the irregularity of sampling positions makes direct modeling challenging. To overcome this, operations such as stacked convolutions, RFAConv, or reshape-Conv can be employed to project the feature maps into higher-dimensional spaces, followed by dimensionality reduction using convolution, thereby approximating the extraction of irregular sampling features.

#### 3.5.3. Extended AKConv

We designed various initial sampling shapes for convolutions of size 5. Therefore, even without using the offset idea in deformable convolutions, AKConv can still generate a wide variety of convolution kernel shapes. But in fact, the size of AKConv can be arbitrary and is not limited to 5. As the size increases, the initial convolution sampling shapes of AKConv become more diverse and rich. Given that target shapes vary across different datasets, designing convolution operations corresponding to the sampling shapes is crucial.

Existing methods, such as deformable convolutions and DSConv, which aim to address the limitations of regular convolutions or are designed for specific target shapes, have not explored convolutional operations for arbitrary sizes and arbitrary sampling shapes. AKConv’s design addresses these limitations by enabling convolutional operations to efficiently extract features from irregular sampling shapes using offsets and by granting convolutional kernels the ability to have an arbitrary number of parameters and multiple shapes.

### 3.6. Context-Mixing Dynamic Convolution

To enable the model to achieve dynamic global modeling capabilities comparable to those of Transformer- and Mamba-based models while retaining strong inductive bias [49], as shown in Figure 4, we introduce a novel context-mixing dynamic convolution (ContMix) that can dynamically model long-range dependencies while maintaining strong inductive bias. Our core idea is to represent the relationship between a token and its context using a set of affinity values between the token and all tokens in a group of region centers within the feature map. These affinity values can then be aggregated in a learnable manner to define a token-level dynamic convolution kernel, thereby injecting contextual knowledge into each weight of the convolution kernel. Once this dynamic kernel is applied to the feature map via a sliding window, each token in the feature map is modulated by the approximate global information collected through the region centers.

### 3.7. Introduction of Wise-IoU

To address the issue of low-quality samples degrading model generalization, this work introduces Wise-IoU [17], a bounding-box regression loss that integrates a dynamic non-monotonic focusing mechanism with a metric-based dual attention strategy [50]. The Intersection over Union (IoU) is a metric used to quantify the overlap between predicted and ground-truth bounding boxes in detection tasks. Here, $W_{i}$ and $H_{i}$ denote the width and height of the overlapping region, while $S_{u}$ represents the union area. During backpropagation, the gradient of $L_{IoU}$ is computed as shown in Equations (Equation 3) and (Equation 4). When no overlap exists between boxes, i.e., $W_{i}=0$ or $H_{i}=0$, the gradient of $L_{IoU}$ vanishes, making the width untrainable.(3)$$\begin{array}{c} L_{IoU}=1-IoU=1-\frac{W_{i}\times H_{i}}{S_{u}} \end{array}$$(4)$$\begin{array}{c} \frac{\partial L_{IoU}}{\partial W_{i}}=\left\{\begin{array}{cc} -H_{i}\times \frac{IoU+1}{S_{u}}, & W_{i}>0\\ 0, & W_{i}=0 \end{array}\right. \end{array}$$

As shown in Equations (Equation 5) and (Equation 6), Wise-IoU introduces a penalty term ($R_{WIoU}$) by separately incorporating $W_{g}$ and $H_{g}$. This enhances the loss ($L_{IoU}$) for normal-quality anchors while avoiding gradient hindrance from $R_{WIoU}$. Such a design weakens geometric penalties under good overlap but prevents excessive interference during training, thereby improving the model’s generalization capability.(5)$$\begin{array}{c} L_{i}=L_{IoU}+R_{i} \end{array}$$(6)$$\begin{array}{c} R_{WIoU}=\exp\left[\frac{{\left(x-x_{gt}\right)}^{2}+{\left(y-y_{gt}\right)}^{2}}{{\left(W_{g}^{2}+H_{g}^{2}\right)}^{2}}\right],\;R_{WIoU}\in \left[1,e\right) \end{array}$$

Furthermore, to reinforce bounding-box regression and mitigate harmful gradients from low-quality samples, Wise-IoU introduces an anchor quality assessment term (β), as defined in Equation (Equation 7). This adjustment assigns smaller gradient increments to outlier anchors. Finally, the Wise-IoU formulation in Equation (Equation 8) integrates both the non-monotonic focusing factor and a dual attention mechanism emphasizing spatial distance.(7)$$\begin{array}{c} \beta =\frac{L_{IoU}^{*}}{\bar{L_{IoU}}},\;\beta \in \left[0,+\infty \right); \end{array}$$(8)$$\begin{array}{c} L_{WIoU}=r\times R_{WIoU}\times L_{IoU},\;r=\frac{\beta }{\delta \times \alpha^{\beta -\delta }}, \end{array}$$

## 4. Experimental Results

### 4.1. Datasets Preparation

This study first selected the white-light imaging (WLI) subset from the PolypDB dataset [51] (as shown in Table 1) as the basis for evaluation. PolypDB is a high-quality medical image database specifically constructed for colorectal polyps.

Its images are sourced from a variety of clinical scenarios and have high annotation consistency and representativeness, providing a reliable benchmark for preliminary model verification.

To overcome the limitations of single datasets in terms of image diversity, sample scale, and clinical coverage scope and to enhance the comprehensiveness and generalizability of model evaluation, this study further introduces two widely used public datasets: Kvasir-SEG [52] and CVC-ClinicDB [53] (as shown in Table 2). Kvasir-SEG, with its rich image count, diverse lesion morphologies, and fine annotations, serves as an authoritative benchmark for evaluating segmentation algorithms, effectively testing the model’s boundary recognition and fine segmentation capabilities for irregular targets under complex endoscopic conditions. Meanwhile, CVC-ClinicDB, derived from different devices and clinical environments, exhibits significant background differences, illumination variations, and quality fluctuations, which can enhance the heterogeneity of test samples and validate the model’s robustness and stability under practical interference scenarios such as low contrast, noise, and blur.

For each dataset, we followed a stratified random sampling strategy to split the data into training, validation, and test sets in an approximate ratio of 8:1:1. Stratification ensures that the distribution of polyp size, morphology, and imaging conditions is consistent across subsets. To minimize labeling noise and batch effects caused by differences in imaging devices and acquisition years, we performed visual quality inspection on all annotations and applied histogram-based intensity normalization across datasets.

By integrating three datasets—namely, PolypDB, Kvasir-SEG, and CVC-ClinicDB—this study constructs a multi-level, multi-dimensional evaluation framework that systematically examines the model’s segmentation performance under different scales, shapes, textures, and imaging conditions, comprehensively demonstrating its cross-domain transfer and generalization capabilities.

### 4.2. Preparation for the Experiment

#### 4.2.1. Experimental Environment

To ensure consistency in the experimental environment and robustness of the methods, all experiments were strictly conducted on a designated server. The server has detailed specifications, as shown in Table 3. The system is equipped with an RTX 4090D graphics processing unit with 24 GB of video memory and 60 GB of memory and is powered by an AMD EPYC 9754 128-core processor with 18 virtual cores. Hard disk storage includes a 30 GB system disk and a 50 GB data disk.

#### 4.2.2. Model Training Strategy

To optimize model performance and enhance generalization capability, a systematic training strategy was employed in this study. Regarding the choice of optimizer, we adopted the stochastic gradient descent (SGD) algorithm, with its parameter configuration presented in Table 4. This strategy effectively ensures convergence stability during the early stages of training. In terms of data augmentation, we designed a multi-level augmentation pipeline that includes two main categories—geometric transformations and color-space perturbations—as detailed in Table 5. By integrating semantic information from diverse samples, the model’s feature discrimination capability is significantly enhanced. This comprehensive augmentation strategy substantially improves the model’s adaptability to variations in illumination, scale differences, and morphological diversity in wireless optical imaging.

#### 4.2.3. Training Parameters

The configuration of hyperparameters is critical to the performance of the YOLOv12-SEG model during optimization. To rigorously evaluate the effectiveness of algorithmic improvements, it is essential to maintain highly consistent hyperparameters before and after any modifications; otherwise, it is impossible to determine whether performance gains stem from the algorithm itself or parameter adjustments. Therefore, this study adopts a standardized set of hyperparameters (as shown in Table 6) to ensure a fair comparison.

We initially performed a grid search on a subset of the PolypDB dataset to identify the optimal initial parameters, which were subsequently validated on two additional datasets. The results demonstrate that the configuration (learning rate: 0.01; batch size: 16; epochs: 200; input size: 640 × 640) achieved stable IoU convergence across all datasets (as shown in Figure 5), confirming its robustness and generalization capability. Ultimately, this unified hyperparameter setup ensured that all experiments were conducted under identical conditions, thereby guaranteeing the fairness and credibility of the performance comparison.

#### 4.2.4. Evaluation Metrics

To evaluate the performance of GDCA-Net, we employed several widely used metrics in polyp segmentation and object detection tasks, including Precision (Pre), Recall (Rec), F1-score (F1), and mean Average Precision (mAP) at different IoU thresholds (mAP@0.5 and mAP@[0.5:0.95]). Let TP, FP, and FN denote the numbers of true positives, false positives, and false negatives, respectively. The metrics are defined as Equations (Equation 9)–(Equation 12): (9)$$\begin{array}{c} \mathrm{Precision}=\frac{\left|TP\right|}{\left|TP|+|FP\right|} \end{array}$$(10)$$\begin{array}{c} \mathrm{Recall}=\frac{\left|TP\right|}{\left|TP|+|FN\right|} \end{array}$$(11)$$\begin{array}{c} \mathrm{mAP}=\frac{1}{m}\sum_{i=1}^{m}{\mathrm{AP}}_{i} \end{array}$$(12)$$\begin{array}{c} F1\;\mathrm{Score}=2\times \frac{\mathrm{Precision}\times \mathrm{Recall}}{\mathrm{Precision}+\mathrm{Recall}} \end{array}$$
where $AP_{i}$ denotes the average precision of the *i*-th class and *N* is the total number of classes. mAP@0.5 represents the mean average precision at an IoU threshold of 0.5, while mAP@[0.5:0.95] indicates the averaged precision across multiple IoU thresholds ranging from 0.5 to 0.95.

In summary, these evaluation metrics form a comprehensive evaluation framework that provides strong support for in-depth analysis of the performance of polyp segmentation models in terms of pixel-level prediction accuracy and the completeness of segmentation results. Precision, recall, F1 score, mAP50, and mAP50-95 are widely recognized as standard benchmarks in YOLO-based segmentation research. These metrics are sufficient to comprehensively evaluate the model’s performance in terms of detection accuracy, lesion sensitivity, and segmentation quality. Although additional measures, such as sensitivity, specificity, accuracy, and AUC, could also be considered, they are mathematically correlated with the metrics already reported. Therefore, we focus on these widely accepted indicators in this study.

### 4.3. Experimental Analysis

#### 4.3.1. Quantitative Comparison and Evaluation

The quantitative evaluation reported in this paper aims to comprehensively examine the effectiveness and generalization ability of the proposed method, GDCA-Net. We used Equations (Equation 9)–(Equation 12) to calculate performance metrics such as the accuracy, recall rate, mAP50, and mAP50-95. Given the diverse instances in the dataset—covering varying lesion sizes, shapes, and textures, as well as differences in endoscopic image quality—this study systematically tested multiple different deep learning models.

This paper focuses on utilizing deep learning models for polyp segmentation, particularly emphasizing early, precise segmentation of polyps to assist in clinical diagnosis. After thoroughly evaluating the dataset, we selected YOLOv12-seg as the primary framework due to its outstanding performance and efficiency in rapidly segmenting polyps of varying sizes and shapes. The segmentation model built on YOLOv12-seg demonstrated significant improvements across multiple performance metrics.

To systematically assess the effectiveness of the proposed method, this study conducted a comprehensive comparative analysis of a series of deep learning-based segmentation techniques and their improvements [43]. These comparison models include YOLOv6-seg [54], YOLOv8-seg, YOLOv8p2-seg [55], YOLOv10n-seg [56], YOLOv11-seg [57], YOLOv12-seg [58], EfficientNetv2-seg [59], vanillanet-seg [60], and ADNet-seg. Additionally, to assess the model’s generalization ability across diverse scenarios and requirements, we selected multiple datasets for experimentation. These include the PolypDB dataset, the Kvasir-SEG dataset, and the CVC-ClinicDB dataset. The comparison results of various performance metrics across different datasets are compiled in Table 7 and Figure 6, comprehensively demonstrating the advantages and disadvantages of the proposed GDCA-Net model compared to other models.

As shown in Table 7, the proposed GDCA-Net model performs well on most core metrics. On the highly challenging PolypDB dataset, GDCA-Net achieved the best results in terms of both the mAP50 and mAP50-95 metrics, with values of 85.9% and 46.9%, respectively. This indicates that the model demonstrates strong robustness when faced with challenging data such as low image quality, uneven lighting, and blurred polyp boundaries. On the high-quality Kvasir-SEG dataset, GDCA-Net also performed exceptionally well. GDCA-Net topped the rankings, with an F1 score of 94.9%, and achieved outstanding results of 97.0% and 74.1% on mAP50 and mAP50-95, respectively. It is worth noting that other advanced models in this dataset, such as YOLOv11-seg and YOLOv8-seg, also achieved very high scores, indicating that the dataset is relatively less challenging. However, GDCA-Net maintains its lead on this high-standard dataset, further validating its advanced capabilities. On the CVC-ClinicDB dataset, GDCA-Net achieved an mAP50 of 98.5% and an mAP50-95 of 82.9%, with performance comparable to that of advanced models such as ADNet-seg and YOLOv8-p2-seg.

Although GDCA-Net does not achieve the highest precision or recall across all datasets when compared to certain models (e.g., YOLOv8-seg and ADNet-seg), it demonstrates consistently high performance across all metrics and datasets, indicating superior overall generalization capability. Those models with marginally higher precision often exhibit a corresponding decrease in recall, highlighting the inherent trade-off between detection sensitivity and false-positive suppression. In contrast, GDCA-Net achieves a more balanced performance profile, which is particularly crucial in clinical applications where both high sensitivity and high specificity are equally critical.

These experimental results strongly demonstrate the effectiveness and robustness of GDCA-Net in handling datasets of different styles and with differing challenges. It is worth noting that in multiple comparison experiments, due to the strict requirements of EfficientNetv2-seg and vanillanet-seg on the dataset, they performed poorly on the PolypDB dataset, resulting in underfitting.

#### 4.3.2. Ablation Experiments

To clarify the contributions and functions of each component in the GDCA-Net model, a series of ablation experiments was conducted in this study. Specifically, eight ablation experiments were performed using the PolypDB dataset, covering the YOLOv12-seg model, models ➀ to ➇, and GDCA-Net. The experimental results are detailed in Table 8. By systematically introducing improved modules based on the YOLOv12-seg baseline model, this study evaluated the roles of the GD mechanism, AKConv, CAF, ContMix, and Wise-IoU. The experimental results clearly demonstrate that the model’s outstanding performance is not accidental but the result of the synergistic effects of multiple key technologies.

In the evaluation of individual components, model ➃ achieved the most significant performance improvement. Its mAP50 improved significantly from 83.7% in the baseline model to 86.4%, and its F1 score also jumped from 84.8% to 86.7%. This significant improvement demonstrates that Wise-IoU can achieve extraordinary effectiveness in optimizing boundary regression and handling complex, uneven segmentation samples. Thus, it has become the core driving force behind the model’s performance improvement. Meanwhile, model ➁ improved the mAP50 by 1.1% and mAP50-95 by 2.1%, demonstrating the effectiveness of this mechanism in enhancing multi-scale feature fusion; model ➂ had a positive impact on recall and mAP50-95, indicating that CAF and ContMix effectively enhance the model’s ability to capture contextual information and irregular morphological features. By integrating global contextual attention and dynamic convolutions that can adapt to irregular shapes, the model can more accurately distinguish polyps in complex backgrounds and precisely segment their diverse, non-linear shapes.

Further experiments reveal the strong synergistic effects between components. When the GD mechanism and AKconv are combined with CAF and ContMix (Model ➄), the model’s performance is further improved in terms of mAP50 and F1 scores. Notably, the combination of CAF and ContMix with Wise-IoU (Model ➅) performs well, achieving an mAP50 of 85.5% and an F1 score of 85.9%, demonstrating strong synergistic gains. The combination of the GD mechanism with AKConv and Wise-IoU (Model ➆) also achieved outstanding performance, with an mAP50 of 85.2% and mAP50-95 of 47.5%.

Finally, by integrating all improved components—GD mechanism with AKConv, CAF with ContMix, and Wise-IoU—into GDCA-Net—the model achieves the best overall performance among all combinations. Although its F1 score of 85.5% is slightly lower than the peak value achieved by Wise-IoU alone, its mAP50 and mAP50-95 reach 85.9% and 46.9%, respectively, approaching optimal levels across all evaluation metrics. This fully demonstrates that GDCA-Net achieves optimal balance and optimization across all performance dimensions by integrating all components, making it a robust model that performs exceptionally well in various complex scenarios.

It should be noted that Models ➃, ➆, and ➇ achieved comparable results, as they share several key components (e.g., Wise-IoU and GD + AKConv), which significantly enhances segmentation performance. However, Model ➇ consistently demonstrates more stable performance across all metrics, indicating that the synergistic integration of modules yields more robust and balanced performance improvements than any individual component alone.

#### 4.3.3. Qualitative Analysis

To comprehensively evaluate the segmentation performance of the GDCA-Net model, this study not only conducted the aforementioned quantitative analysis but also performed qualitative analysis. First, we randomly selected 12 images from the PolypDB dataset [51], as shown in Figure 7a. The GDCA-Net demonstrated strong robustness and adaptability under challenging conditions such as low image quality, uneven lighting, complex backgrounds, and blurred polyp boundaries. The model can reliably segment polyps of various shapes and handles blurred boundaries with great precision, with its prediction results highly consistent with the ground-truth labels (Figure 7b).

Second, to assess the model’s generalization ability, we randomly selected 12 images from the Kvasir-SEG dataset [52], as shown in Figure 7c. The images in this dataset have relatively high quality, with polyp boundaries typically being clear. GDCA-Net also demonstrated exceptional segmentation capabilities on this dataset, with its prediction results matching the true labels in Figure 7d. This demonstrates that GDCA-Net can effectively utilize the rich information in high-quality images to achieve high-precision segmentation and successfully generalize to datasets of different styles.

In addition, to comprehensively evaluate the segmentation performance of GDCA-Net across different clinical scenarios, we randomly selected 12 images from the CVC-ClinicDB dataset, as shown in Figure 7e. Polyps in this dataset are often characterized by uneven illumination, mucus interference, and complex surrounding tissue textures. Under these challenging conditions, GDCA-Net still demonstrates excellent segmentation performance, accurately capturing both the overall structure and subtle contours of polyps. Its predicted results show a high degree of consistency with the ground-truth labels in Figure 7f; even in the presence of slight occlusion, the model maintains stable segmentation consistency.

Overall, the GDCA-Net model proposed in this study performs well in tasks involving different image qualities and polyp features, covering a wide range of scenarios, including blurry, complex, clear, and simple ones. On the PolypDB core dataset, GDCA-Net can accurately and effectively segment various types of polyps, thereby providing important auxiliary support for clinical doctors in early diagnosis.

#### 4.3.4. Failure Cases Analysis

Although GDCA-Net demonstrates superior segmentation performance across various datasets, we also observed a few typical failure cases during qualitative analysis, as shown in Figure 8. Including and analyzing these cases is essential for understanding the current limitations of the proposed method and guiding future improvements. As shown in Figure 8, GDCA-Net occasionally fails to detect polyps with extremely low contrast, smooth texture, or unclear boundaries, particularly when they are small or flat against the surrounding mucosa. In these cases, the model struggles to differentiate subtle intensity variations, leading to incomplete or missed segmentation regions. In addition, the model sometimes misclassifies bright reflections caused by endoscopic illumination as polyp regions. These false positives are likely due to the similar intensity distribution between specular highlights and actual lesions, which confuses the feature extraction process.

These failure cases highlight potential areas for improvement. Future work will focus on incorporating illumination-invariant feature representations and context-aware refinement mechanisms to better handle challenging imaging conditions and reduce misclassification.

## 5. Conclusions

This study proposes an improved gastrointestinal polyp segmentation algorithm based on an enhanced version of YOLOv12-seg. Compared to the base model, the proposed algorithm framework first reconstructs the neck network by introducing the GD mechanism, significantly enhancing multi-scale feature fusion capabilities; second, it employs the AKConv module to achieve dynamic sampling of arbitrarily shaped convolution kernels, enhancing the modeling capability of irregularly shaped polyp edges; Furthermore, it integrates the CAFM and ContMix modules to combine local details with global contextual information, optimizing the modeling of long-range dependencies in segmentation boundaries; finally, it introduces the Wise-IoU loss function, utilizing a dynamic non-monotonic focusing mechanism and mask quality evaluation strategy to enhance the model’s robustness against low-quality samples. Empirical results show that this algorithm outperforms existing methods in terms of polyp segmentation accuracy and boundary consistency, achieving significant results on both public datasets and internal clinical data while maintaining real-time inference efficiency.

Although the algorithm performs exceptionally well, it still exhibits minor segmentation errors under extreme lighting conditions, small polyps, and blurred boundaries. Future work will focus on the following directions: constructing a multi-center, multi-modal polyp image dataset to deeply analyze the correlation between polyp morphological features and segmentation accuracy; exploring lightweight architectures based on Transformers to balance model complexity and real-time segmentation requirements; and combining pathological prior knowledge to study the mapping relationship between polyp malignancy and segmentation boundaries, providing more comprehensive decision support for clinical diagnosis. The ultimate goal is to achieve a high-precision, robust intelligent segmentation system through algorithm optimization and multidisciplinary research.

## Figures and Tables

**Figure 1 sensors-25-06495-f001:**
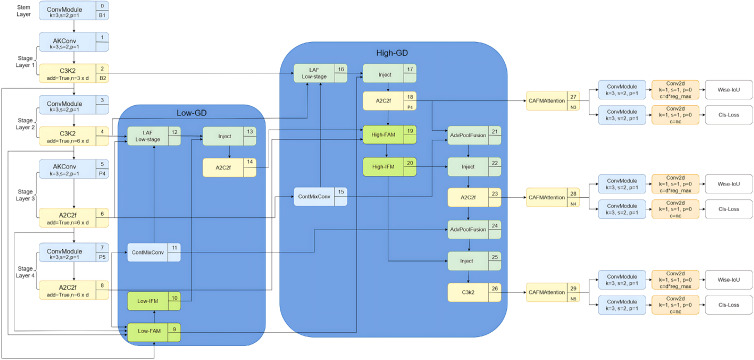
Improved network architecture. The Gather-and-Distribute (GD) mechanism is integrated into the neck of the original You Only Look Once version 12 segmentation model (YOLOv12-Seg) network: Low-GD replaces the up-sampling fusion in Path Aggregation Network (PANet), while Higher-GD substitutes the down-sampling fusion.

**Figure 2 sensors-25-06495-f002:**
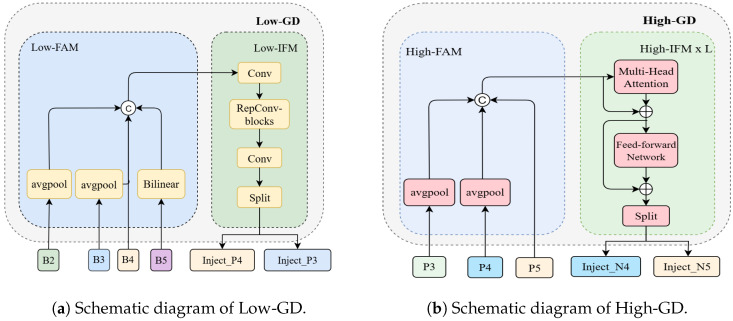
Structure of the GD mechanism. The GD mechanism enhances the model’s ability to detect objects of varying sizes by constructing two dedicated branches: the Low-Stage Gather-and-Distribute Branch (Low-GD) and the High-Stage Gather-and-Distribute Branch (High-GD).

**Figure 3 sensors-25-06495-f003:**
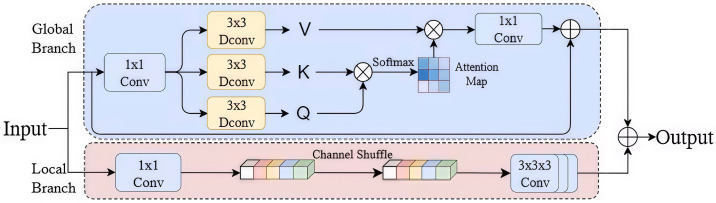
Module diagram of the attention and convolution fusion between the local branch and the global branch. The former specializes in efficiently extracting local details and facilitating inter-channel interactions, while the latter focuses on modeling long-range feature dependencies and capturing global spatial relationships.

**Figure 4 sensors-25-06495-f004:**
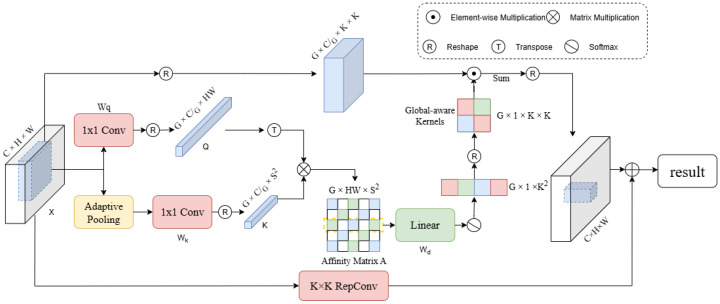
Dynamic convolution structure diagram. The advantage of Alterable Kernel Convolution (AKConv) lies in its ability to efficiently model complex kernel shapes by learning offsets and dynamic sampling grids while maintaining flexible sampling.

**Figure 5 sensors-25-06495-f005:**
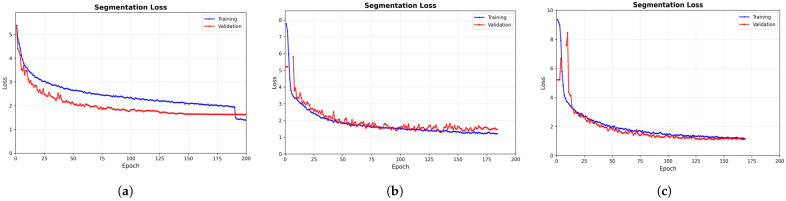
Segmentation loss curves during training. Blue: training; red: validation. Both losses converge smoothly, indicating effective model training. Note that training on the Kvasir-SEG and CVC-ClinicDB datasets was terminated around epoch 175 due to the early stopping criterion. (**a**) Loss trend observed on the PolypDB dataset. (**b**) Loss trend observed on the Kvasir-SEG dataset. (**c**) Loss trend observed on the CVC-ClinicDB dataset.

**Figure 6 sensors-25-06495-f006:**
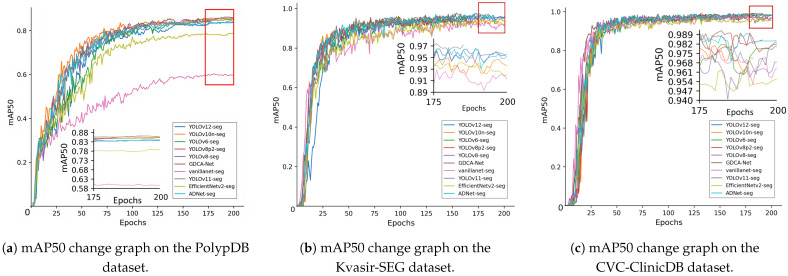
Curves of model evaluation metrics. (**a**) The mean Average Precision at 50% IoU threshold (mAP50) metric change graph of GDCA-Net under the PolypDB dataset. (**b**) The mAP50 metric change graph of GDCA-Net under the Kvasir-SEG dataset. (**c**) The mAP50 metric change graph of GDCA-Net under the CVC-ClinicDB dataset.

**Figure 7 sensors-25-06495-f007:**
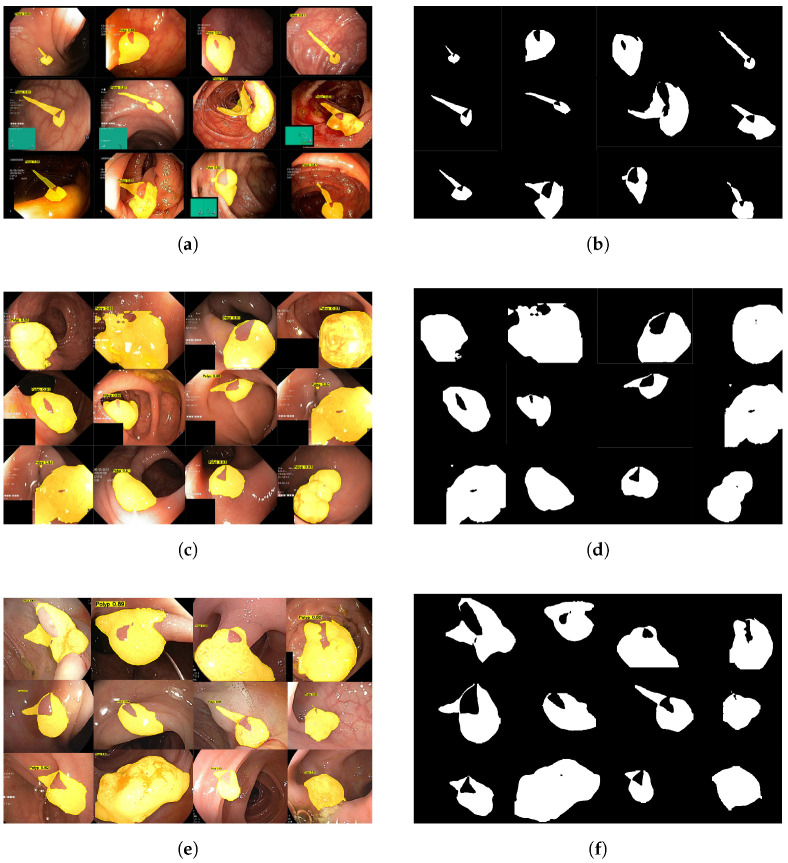
Qualitative comparisons between the segmentation results of GDCA-Net and the ground truth on samples from three datasets. From left to right: segmentation outputs of GDCA-Net and ground-truth masks. The results are evaluated based on the Intersection over Union (IoU) metric, where a higher IoU indicates a closer match between the predicted mask and the ground truth. (**a**) Segmentation results on the PolypDB dataset. (**b**) The mask on the PolypDB dataset. (**c**) Segmentation results on the Kvasir-SEG dataset. (**d**) The mask on the Kvasir-SEG dataset. (**e**) Segmentation results on the CVC-ClinicDB dataset. (**f**) The mask on the CVC-ClinicDB dataset.

**Figure 8 sensors-25-06495-f008:**
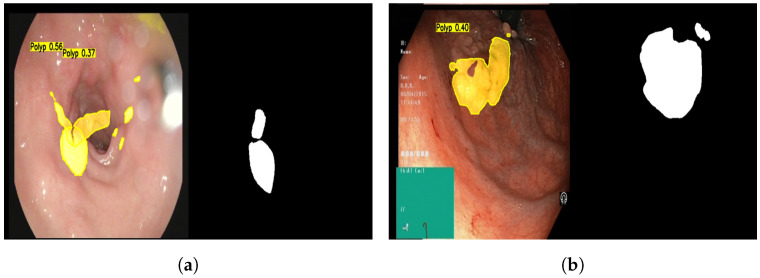
Examples of failure cases encountered by GDCA-Net. These cases reveal potential areas for improvement in illumination invariance and contextual understanding. (**a**) False positives caused by specular highlights misclassified as polyp regions. (**b**) Missed detection of flat polyps with low contrast and unclear boundaries.

**Table 1 sensors-25-06495-t001:** Description of WLI from the PolypDB dataset.

Source	Category	Sample Count
High-quality colonoscopic images from multiple medical centers worldwide [51]	Training set	2870
	Validation set	358
	Test set	360
	Total	3588

**Table 2 sensors-25-06495-t002:** Kvasir-SEG and CVC-ClinicDB dataset description.

Dataset	Source	Training Set	Validation Set	Test Set
Kvasir-SEG	Vestre Viken Health Trust [52]	800	100	100
CVC-ClinicDB	Hospital Clinic, Barcelona and Computer Vision Center [53]	489	61	62

**Table 3 sensors-25-06495-t003:** Description of the software and hardware environment.

Device Type	Device Description
GPU	RTX 4090D 24 GB
CPU	18 vCPU AMD EPYC 9754 128-Core Processor (AMD)
RAM	60 GB
Storage	System disk: 30 GB; Data disk: 50 GB

**Table 4 sensors-25-06495-t004:** Optimizer parameter configuration.

Parameter Category	Parameter Name	Setting Value
Optimizer	Type	SGD
Basic Parameters	Initial Learning Rate	0.01
	Momentum	0.937
	Weight Decay	0.0005
Warm-up Strategy	Warm-up Period	3
	Warm-up Momentum	0.8

**Table 5 sensors-25-06495-t005:** Data Augmentation Strategy.

Augmentation Type	Specific Method	Parameter Setting
Geometric Transformation	Random Horizontal Flip	Probability: 0.5
	Random Translation	Ratio: 0.1
	Random Scaling	Ratio: 0.5
Color Augmentation	Random Erasing	Probability: 0.4
	AutoAugment	randaugment
Advanced Augmentation	Copy–Paste	Probability: 0.1; Mode: Flip

**Table 6 sensors-25-06495-t006:** Training hyperparameter settings.

Dataset	Training Hyperparameters	Setting Quantity
PolypDB	Learning rate	0.01
	Batch size	16
	Epoch	200
	Image size	640
Kvasir-SEG	Learning rate	0.01
	Batch size	16
	Epoch	200
	Image size	640
CVC-ClinicDB	Learning rate	0.01
	Batch size	16
	Epoch	200
	Image size	640

**Table 7 sensors-25-06495-t007:** Comparison of experimental results.

Dataset	Model	Precision (%)	Recall (%)	mAP50 (%)	mAP50-95 (%)	F1 Score
PolypDB	YOLOv6-seg	93.4	79.8	84.9	44.8	86.1
	YOLOv8p2-seg	88.5	81.3	84.9	44.9	84.7
	YOLOv8-seg	88.2	82.9	85.3	46.3	85.5
	yolov10n-seg	90.1	82.5	85.4	46.6	86.1
	YOLOv11-seg	91.6	79.4	84.3	45.4	85.1
	YOLOv12-seg	90.5	79.7	83.7	44.0	84.8
	EfficientNetv2-seg	82.8	76.9	78.7	37.5	79.7
	vanillanet-seg	72.0	62.7	59.8	24.9	67.0
	ADNet-seg	89.3	80.7	83.6	44.7	84.8
	GDCA-Net	90.6	81.0	85.9	46.9	85.5
Kvasir-SEG	YOLOv6-seg	89.7	92.0	93.3	71.6	90.8
	YOLOv8p2-seg	91.7	88.7	94.7	73.8	90.2
	YOLOv8-seg	95.8	86.0	96.0	76.0	90.6
	yolov10n-seg	88.2	89.7	94.9	72.9	88.9
	YOLOv11-seg	95.7	90.0	96.9	73.8	92.8
	YOLOv12-seg	94.4	89.0	96.5	74.8	91.6
	EfficientNetv2-seg	91.9	87.0	94.0	65.7	89.4
	vanillanet-seg	87.0	88.0	92.5	67.2	87.5
	ADNet-seg	88.6	93.0	96.3	73.5	90.7
	GDCA-Net	93.9	96.0	97.0	74.1	94.9
CVC-ClinicDB	YOLOv6-seg	98.3	94.8	97.1	82.7	96.5
	YOLOv8p2-seg	98.1	96.7	98.3	83.8	97.4
	YOLOv8-seg	97.3	91.8	96.0	83.2	94.5
	yolov10n-seg	98.3	94.8	97.0	82.8	96.5
	YOLOv11-seg	94.5	95.1	98.4	82.4	94.8
	YOLOv12-seg	83.5	95.1	95.0	81.2	88.9
	EfficientNetv2-seg	96.2	91.8	95.5	75.4	93.9
	vanillanet-seg	98.2	87.4	97.8	81.2	92.5
	ADNet-seg	98.3	94.9	98.4	83.3	96.6
	GDCA-Net	98.1	93.4	98.5	82.9	95.7

**Table 8 sensors-25-06495-t008:** Results of ablation experiments.

Num	GD and AKConv	CAF and ContMix	Wise-IoU	Precision (%)	Recall (%)	mAP50 (%)	mAP50-95 (%)	F1 Score
➀				90.5	79.7	83.7	44.0	84.8
➁	✓			90.3	79.7	84.8	46.1	84.7
➂		✓		87.4	81.0	83.9	44.9	84.1
➃			✓	90.1	83.5	86.4	45.3	86.7
➄	✓	✓		89.8	81.9	84.9	45.6	85.7
➅		✓	✓	90.6	81.7	85.5	46.2	85.9
➆	✓		✓	91.3	78.5	85.2	47.5	84.4
➇	✓	✓	✓	90.6	81.0	85.9	46.9	85.5

## Data Availability

The data presented in this study are available upon request from the corresponding author.

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
