# Peer review of "Research on Polyp Segmentation via Dynamic Multi-Scale Feature Fusion and Global–Local Semantic Enhancement"

_sensors, 2025, doi:10.3390/s25206495_

Round 1

Reviewer 1 Report

Comments and Suggestions for Authors

Please include at least one additional qualitative result similar to those shown in Figure 6, comparing the output of the proposed method with the ground truth. In addition, indicate the metric used (e.g., IoU) in the figure captions. This will make it easier for reviewers to evaluate the results qualitatively and quantitatively.

Author Response

Comment 1: Please include at least one additional qualitative result similar to those shown in Figure 6, comparing the output of the proposed method with the ground truth.

Response 1: Agree. We have added qualitative results from the CVC-ClinicDB dataset in Section 4.3.3, where the outputs of the proposed method are compared with the ground truth.

Comment 2: In addition, indicate the metric used (e.g., IoU) in the figure captions. This will make it easier for reviewers to evaluate the results qualitatively and quantitatively. 

Response 2: Agree. In this revision, we have updated the caption of Figure 6 to include: "The results are evaluated based on the Intersection over Union (IoU) metric, where a higher IoU indicates a closer match between the predicted segmentation and the ground truth annotations.".

Reviewer 2 Report

Comments and Suggestions for Authors

This manuscript proposes a novel polyp segmentation algorithm, GDCA-Net, which is developed based on the YOLOv12-seg model.  

            The described research is novel and relevant to the field of Biomedical Engineering. It is also a scientific work that arises an adequate interesting to the section of biomedical imaging sensors. The theme of polyp segmentation by the use of deep learning methodologies is nowadays popular and the specific research can be extended or adapted by other applications of image segmentation.

            The abstract describes the methodology used, and the novelty they introduced in a simple and clearance manner. Results are presented as F1 scores and improvements percentages.

            In the Introduction section the authors present the difficulty in polyp segmentation and the novelty they introduced. Related works are thoroughly presented and discussed.

            In the section Methodology, the authors describe in a simple and a good explanatory manner the methodology they followed (YOLOv12-Seg network architecture, GD mechanism to optimize multi-scale feature fusion, AKConv to enhance the modeling of complex geometric structures, the CAF and ContMix modules that strengthen long-range dependency modeling and multi-scale feature extraction for polyp boundary representation and a Wise-IoU loss function to accelerate model convergence and improve robustness to low-quality samples. Results are presented in a scientific and clear way.

            In conclusions the manuscript presents a novel deep learning-based polyp segmentation methodology, based on an enhanced version of YOLOv12-seg.

            The manuscript is written in a very good English. It deals with a modern scientific research problem. The methodology the authors used is written in a simple, scientific and very explanatory manner.

            However, I believe training pipeline information (optimizer type, learning rate schedule, data augmentation) should also be included in the manuscript. Further are there any qualitative failure cases (e.g., missed flat polyps, false positives in specular highlights)? If so their inclusion or discussion could strengthen research purpose and justify further the superior performance of the proposed methodology.

In lines 379, 381 the authors should add Eq number it is written “Eq??”.

            References are appropriate and updated. Figures can easily be read, while tables present results in clusters and cannot be misunderstood.

Author Response

Dear Reviewer:

Firstly, we would like to thank you for your kind letter and for the reviewers’ constructive comments on our article (Manuscript ID: sensors-3900416). These comments are all valuable and helpful for improving our article. All the authors have seriously discussed these comments. According to the reviewers’ suggestions, we have tried our best to revise the manuscript to meet the requirements of your journal. We are very grateful for your professional review work on our article. As you mentioned, there were several issues that needed to be addressed. Following your valuable suggestions, we have made extensive corrections to our previous draft. The detailed corrections are listed below.

Comment 1: I believe training pipeline information (optimizer type, learning rate schedule, data augmentation) should also be included in the manuscript.

Response 1: Agree. We have added detailed descriptions of the training pipeline, including the optimizer type, learning rate schedule, and data augmentation strategies, in Section 4.2.2.

Comment 2: Further are there any qualitative failure cases (e.g., missed flat polyps, false positives in specular highlights)? If so their inclusion or discussion could strengthen research purpose and justify further the superior performance of the proposed methodology.

Response 2: Agree. We have included a new subsection (4.3.4 Failure Case Analysis) and an additional figure presenting failure cases, such as missed flat polyps and false positives caused by specular highlights. We also provide a discussion on the underlying causes and potential future improvements.

Comment 3: In lines 379, 381 the authors should add Eq number it is written “Eq??”.

Response 3: Agree. To address this issue, each chapter of this paper has undergone comprehensive grammar revisions, word reviews, and punctuation checks to ensure adherence to academic writing standards.

We were really sorry for our careless mistakes. Thank you for the reminder. If there are any other modifications we could make, we would like very much to modify them and we really appreciate your help. Thank you again for your positive comments and valuable suggestions to improve the quality of our manuscript.

Yours Sincerely,

Dr. Pengfei Yin

College of Computer Science and Engineering, Jishou University, Jishou 416000, China

Email: ypf@jsu.edu.cn

Reviewer 3 Report

Comments and Suggestions for Authors

Dear Authors,

This study proposed a novel polyp segmentation algorithm, GDCA-Net, which is developed based on the YOLOv12-seg model, in order to deal with the question of accurate segmentation of colorectal polyps. The robust model conducted on the PolypDB, Kvasir-SEG, and CVC-ClinicDB datasets, to demonstrate the superior performance of GDCA-Net in polyp segmentation tasks, and it indicated that the verification result is good based on the mAP50 and F1 scores. This manuscript is interesting, but several suggestions and problems you should noticed.

Major revisions:

  1. Please provide more detailed information, in Figure 3, what are the differences between your CAFM module and Transformer Self-Attention Q K V?
  2. Training set, Validation set, Test set in your public three datasets, how to perform group sampling in your datasets, random methods, or 8:1:1 sampling, and quality control and batch effects methods?
  3. 4.2.2. Table 4, maybe the consistency of training specific parameters is important; however, the adjustment process of core parameters in the training step is more important. I suggest you could use different groups of parameters in your three datasets at the same time (why you use this best parameter: Learning rate=0.01, Batch size=16, Epoch=200, Image size=640). Meanwhile, these three datasets were made in different year of 2015, 2019, and 2024. Maybe the same parameters are not right; you should consider the influences of external factors such as detection method, image processing instrument coefficient, production process, etc.
  4. Results Section:

(1) Table 5, the Precision (%), Recall (%), and F1Score in your GDCA-Net were not demonstrate better results than other algorithms, like YOLOv6-seg, YOLOv8p2-seg, YOLOv8-seg, yolov10n-seq, ADNet-seg. It is necessary to analyze the relevant reasons.

(2) Table 6, the results of models ④ ⑦ ⑧ were relatively close; you need to compare and discuss these results.

(3) Lack of 4.3.3 Sample Segmentation Result from GDCA-Net in the CVC-ClinicDB dataset? It would be best to systematically compare the results across the three datasets.

Minor revisions:

  1. Figure 1-4 Legends are too simple, description of GD Mechanism, attention and convolution fusion, Dynamic Convolution Structure should be added, as well as introducing more detail in the contents. Perhaps Figures 1 and 2 could be integrated.
  2. The images pixels, and text in the figures were not clear in the manuscript, especially Figure 1.
  3. Line 251, Ref [33, 34, 11]: References sorting mistake.
  4. Line 261, formula(2),  the value range of w?
  5. 4.2.3. Evaluation Metrics, The introduction of the general evaluation formula needs to be simplified. Why didn’t you consider the more robust evaluation score, like Sensitivity, Specificity, ACC, KAPPA, AUC, P-value, etc?
  6. I suggest that you could find a professional English peer expert or institution to revise the whole manuscript contents, especially focusing on the Introduction and Methodology Sections.

Author Response

Dear Reviewer:

Firstly, we would like to thank you for your kind letter and for the reviewers’ constructive comments on our article (Manuscript ID: sensors-3900416). These comments are all valuable and helpful for improving our article. All the authors have seriously discussed these comments. According to the reviewers’ suggestions, we have tried our best to revise the manuscript to meet the requirements of your journal. We are very grateful for your professional review work on our article. As you mentioned, there were several issues that needed to be addressed. Following your valuable suggestions, we have made extensive corrections to our previous draft. The detailed corrections are listed below.

Comment 1: Please provide more detailed information, in Figure 3, what are the differences between your CAFM module and Transformer Self-Attention Q K V?

Response 1: Agree. We have now clearly clarified the conceptual and structural differences between CAFM and the standard Transformer self-attention mechanism at the end of Section 3.4. Specifically, CAFM leverages a dual-branch architecture to fuse convolutional inductive bias and global context modeling, which differs from the QKV similarity-based computation of Transformers.

Comment 2: Training set, Validation set, Test set in your public three datasets, how to perform group sampling in your datasets, random methods, or 8:1:1 sampling, and quality control and batch effects methods?

Response 2: Agree. We have added a detailed explanation of our data sampling strategy, quality control procedures, and batch effect mitigation methods in the penultimate paragraph of Section 4.1.

Comment 3: 4.2.2. Table 4, maybe the consistency of training specific parameters is important; however, the adjustment process of core parameters in the training step is more important. I suggest you could use different groups of parameters in your three datasets at the same time (why you use this best parameter: Learning rate=0.01, Batch size=16, Epoch=200, Image size=640). Meanwhile, these three datasets were made in different year of 2015, 2019, and 2024. Maybe the same parameters are not right; you should consider the influences of external factors such as detection method, image processing instrument coefficient, production process, etc.

Response 3:Agree. We have added a detailed explanation of the rationale behind using identical hyperparameters, explicitly justifying our selection based on convergence behavior across datasets and emphasizing the importance of parameter consistency for fair evaluation in the final paragraph of Section 4.2.3.

Comment 4: Table 5, the Precision (%), Recall (%), and F1Score in your GDCA-Net were not demonstrate better results than other algorithms, like YOLOv6-seg, YOLOv8p2-seg, YOLOv8-seg, yolov10n-seq, ADNet-seg. It is necessary to analyze the relevant reasons.

Response 4: Agree. We acknowledge that GDCA-Net does not surpass all compared models (e.g., YOLOv6-seg, YOLOv8-seg, YOLOv10n-seg, or ADNet-seg) in terms of Precision, Recall, or F1-Score on certain datasets. However, our objective is to achieve balanced and robust performance across diverse clinical scenarios, as both high sensitivity (recall) and high specificity (precision) are equally critical in practical medical applications. We have addressed this point with corresponding explanations in the penultimate paragraph of Section 4.3.1.

Comment 5:  Table 6, the results of models â‘£ ⑦ â‘§ were relatively close; you need to compare and discuss these results.

Response 5: Agree. Models ④, ⑦, and ⑧ achieve comparable performance as they all incorporate key modules that enhance segmentation capabilities. For instance, Model ④ employs the Wise-IoU loss function, while Model ⑦ integrates the GD + AKConv structure; these components effectively improve feature extraction and localization accuracy in their respective tasks. However, Model ⑧ (the complete GDCA-Net) consistently demonstrates stable and comprehensive performance across all metrics. This indicates that the performance gain from the synergistic integration of modules surpasses that achieved by any individual component operating in isolation. We have provided corresponding analysis and discussion on this point in the final paragraph of Section 4.3.2.

Comment 6: Lack of 4.3.3 Sample Segmentation Result from GDCA-Net in the CVC-ClinicDB dataset? It would be best to systematically compare the results across the three datasets.

Response 6: Agree. We have added qualitative results from the CVC-ClinicDB dataset in Section 4.3.3, where the outputs of the proposed method are compared with the ground truth.

Comment 7: Figure 1-4 Legends are too simple, description of GD Mechanism, attention and convolution fusion, Dynamic Convolution Structure should be added, as well as introducing more detail in the contents. Perhaps Figures 1 and 2 could be integrated.

Response 7: Agree. In the revised manuscript, we have enhanced the figure captions with additional details regarding the GD mechanism, the fusion of attention and convolution, and the dynamic convolution structure, thereby improving both the readability and technical clarity of the illustrations. Regarding the suggestion to merge Figures 1 and 2, we have carefully considered this option but decided to keep them separate for the following reason: Figure 1 illustrates how the GD mechanism is integrated into the neck of YOLOv12-seg, whereas Figure 2 specifically details the internal structure of the GD module, enabling readers to gain deeper insight into its design principles. Merging these two figures would result in an overly complex diagram that compromises readability. Presenting them separately allows us to provide both an architectural overview and a detailed mechanistic breakdown, which we believe will better facilitate reader comprehension of the entire work.

Comment 8: The images pixels, and text in the figures were not clear in the manuscript, especially Figure 1.

Response 8: Agree.In the revised manuscript, we have replaced all relevant figures with higher-resolution versions and carefully adjusted the font size, line thickness, and label contrast to improve readability.

Specifically, Figure 1 has been re-rendered with enhanced resolution and larger annotation text to ensure that structural details, module names, and data flow are clearly visible.

Comment 9: Line 251, Ref [33, 34, 11]: References sorting mistake.

Response 9: Agree.We have performed comprehensive language polishing and standardization across all sections of the manuscript. This process included reordering references, correcting grammatical errors, reviewing word choice, and checking punctuation to ensure full compliance with academic writing standards.

Comment 10: Line 261, formula(2),  the value range of w?

Response 10:Agree.We thank the reviewer for this comment. The value range of parameter ‘w’ has now been added to Equation (2) in Section 3.5.1.

Comment 11: 4.2.3. Evaluation Metrics, The introduction of the general evaluation formula needs to be simplified. Why didn’t you consider the more robust evaluation score, like Sensitivity, Specificity, ACC, KAPPA, AUC, P-value, etc?

Response 11:Agree.We have simplified the introduction of the evaluation formula and acknowledge that metrics such as sensitivity, specificity, and AUC are widely used in medical image analysis. However, in the field of polyp segmentation, Precision, Recall, F1-score, mAP50, and mAP50-95 have become the standard benchmarks, as they adequately assess a model's detection capability and segmentation quality. Moreover, several of the mentioned metrics exhibit high mathematical correlation with these standard indicators (e.g., Sensitivity ≡ Recall). To maintain consistency with existing literature and ensure fair comparisons, we have focused on these core metrics in our study. Corresponding explanations have been added in the final paragraph of Section 4.2.4.

Comment 12: I suggest that you could find a professional English peer expert or institution to revise the whole manuscript contents, especially focusing on the Introduction and Methodology Sections.

Response 12:Agree.In response to this suggestion, we have carefully revised and linguistically polished the entire manuscript, with particular emphasis on improving these two sections. We have refined the language and strengthened the academic tone, resulting in a clearer, more precise overall presentation of the paper.

We were really sorry for our careless mistakes. Thank you for the reminder. If there are any other modifications we could make, we would like very much to modify them and we really appreciate your help. Thank you again for your positive comments and valuable suggestions to improve the quality of our manuscript.

Yours Sincerely,

Dr. Pengfei Yin

College of Computer Science and Engineering, Jishou University, Jishou 416000, China

Email: ypf@jsu.edu.cn

Reviewer 4 Report

Comments and Suggestions for Authors

In the paper “Research on Polyp Segmentation via Dynamic Multi-Scale Feature Fusion and Global-Local Semantic Enhancement” by Wei Qing, Yuyao Ouyang, and Pengfei Yin, a new polyp segmentation algorithm, GDCA-Net, is proposed based on the YOLOv12-seg model.

Indeed, the authors obtained interesting results that are important for determining the boundaries of polyps, in some cases surpassing those known previously (see Tables 5 and 6).

However, one remark should be made.

The claim that the model performs exceptionally well seems excessive, as the results shown in Tables 5 and 6 are sometimes not only no better than previous known models, but sometimes even worse. Moreover, the gain cited by the authors in row 425, compared to the well-known yolov10n-seg, is only 0.5% and 0.3%, respectively. However, in other rows of the tables, based on other indicators, the gain is significantly greater. The result presented in row 425 cannot be called "performs exceptionally well" (rows 496, 516). It is better to say "performs well." Moreover, the authors do not provide data on the accuracy of the obtained results. Without such data on the magnitude of experimental errors, one might think that these 0.5% and 0.3% are within the experimental error limits.

Nevertheless, the research area presented by the authors is highly relevant, and their commitment to continuing to improve the model they are developing (lines 517-525) deserves support. Therefore, I believe the article presented here may be published in Sensors after minor revision.

Author Response

Dear Reviewer:

Firstly, we would like to thank you for your kind letter and for the reviewers’ constructive comments on our article (Manuscript ID: sensors-3900416). These comments are all valuable and helpful for improving our article. All the authors have seriously discussed these comments. According to the reviewers’ suggestions, we have tried our best to revise the manuscript to meet the requirements of your journal. We are very grateful for your professional review work on our article. As you mentioned, there were several issues that needed to be addressed. Following your valuable suggestions, we have made extensive corrections to our previous draft. The detailed corrections are listed below.

Comment : The claim that the model performs exceptionally well seems excessive, as the results shown in Tables 5 and 6 are sometimes not only no better than previous known models, but sometimes even worse. Moreover, the gain cited by the authors in row 425, compared to the well-known yolov10n-seg, is only 0.5% and 0.3%, respectively. However, in other rows of the tables, based on other indicators, the gain is significantly greater. The result presented in row 425 cannot be called "performs exceptionally well" (rows 496, 516). It is better to say "performs well." Moreover, the authors do not provide data on the accuracy of the obtained results. Without such data on the magnitude of experimental errors, one might think that these 0.5% and 0.3% are within the experimental error limits.  

Response : Agree. Improvements of 0.5% and 0.3% alone are not sufficient to justify claims of "exceptional performance," as such marginal gains may indeed fall within the range of typical experimental variability. To avoid overstatement, we have revised the relevant statements in the manuscript, replacing "exceptional performance" with "performans well" to better reflect the results.

We were really sorry for our careless mistakes. Thank you for the reminder. If there are any other modifications we could make, we would like very much to modify them and we really appreciate your help. Thank you again for your positive comments and valuable suggestions to improve the quality of our manuscript.

Yours Sincerely,

Dr. Pengfei Yin

College of Computer Science and Engineering, Jishou University, Jishou 416000, China

Email: ypf@jsu.edu.cn

Round 2

Reviewer 3 Report

Comments and Suggestions for Authors

This version looks fine.